# Feats of Clay: Considering the Materiality of Late Bronze Age Cyprus

**Louise Steel** 

Y Athrofa: Humanities and Education, University of Wales Trinity Saint David, Lampeter, Ceredigion SA48 7ED, UK; l.steel@uwtsd.ac.uk

**Abstract:** This paper examines the materiality of the Cypriot Base Ring ware through the lens of the new materialisms. Specifically, it draws upon Bennett's vibrant matter and thing-power, to explore how cultural and technological knowledges of Late Bronze Age Cyprus were informed through material engagements with clay. This approach highlights the agency of matter and illustrates how the distinct capacities of clay (*working with* water and fire) provoked, enabled and constrained potters' behaviour, resulting in a distinctive pottery style that was central to the Late Cypriot social and material world. The aim is to demonstrate how people, materials and objects are all matter in relationship, drawing attention to the fluidity, porosity and relationality of the material world.

**Keywords:** Cyprus; Late Bronze Age; Ring Base; new materialisms; thing-power; clay

## 1. Introduction

This paper examines the archaeology of Late Bronze Age Cyprus (Late Cypriot period) through a new materialist lens [1], exploring how matter and substances, including humans, come together in material entanglements. This approach firmly positions humans as part of the material world, recognizing how they not only shape but are equally shaped by the matter of the world. The aim is to move away from notions grounded in Enlightenment ontologies that view the matter of the world (including land, water, clay, stone, metals, plants and animals) as an inert resource, waiting to be shaped, transformed and ascribed meaning by people. Instead, it focuses on the very materiality of being human [2], highlighting how people and other materials are in a relationship, which might be explored as assemblage (or *agencement*) [3,4] or otherwise as things-in-phenomena [4,5]. The assemblage approach in particular is gaining traction within archaeological literature [6–8]. The new materialisms [2,9], in particular, highlight the distinct capacities (the agency or vitality) [10] of diverse substances and how these provoked, enabled and constrained human behaviours. Focusing on material engagements and the myriad intersections between human and non-human (earthy) matter, this paper explores the materiality of Late Cypriot communities and how they created and shaped new social and material worlds through their daily encounters with these substances.

## 2. Late Bronze Age Cyprus

The Late Cypriot (LC) period (Table 1) is typically characterized as a period of intense culture contact—as the island was increasingly embedded in long-distance maritime trade networks—resulting in technological and cultural innovation [11–13]. One result of greater contact with the Near East was an apparent shift to urban communities [14,15] (Figure 1), although the material evidence resists archaeologists' attempts to shoehorn the island into models of complex societies and state formation [16,17]. There was greater diversity in landscape use throughout the island than in the preceding Early-Middle Cypriot (EC-MC) periods, resulting in a progressively complex settlement hierarchy, characterized by coastal urban centres engaged in maritime trade and various tiers of smaller

specialist production sites in the hinterland [18,19]. Inevitably these social changes were accompanied by greater human engagement with, and manipulation of, the material world, in particular increased exploitation of copper [20], monumentality within the urban context [21,22], and wealthy burials [23], although our understanding of the power strategies and ideologies developed to control economic resources is limited.

**Table 1.** Chronological Table for Bronze Age Cyprus.

| Cultural Phase | Approximate Date BCE (Calibrated) |
| --- | --- |
| Middle Cypriot III–Late Cypriot I | 1750/1700–1450 |
| Late Cypriot IIA–Late Cypriot IIC early | 1450–1300 |
| Late Cypriot IIC (late)–Late Cypriot IIIA | 1300–1125/1100 |

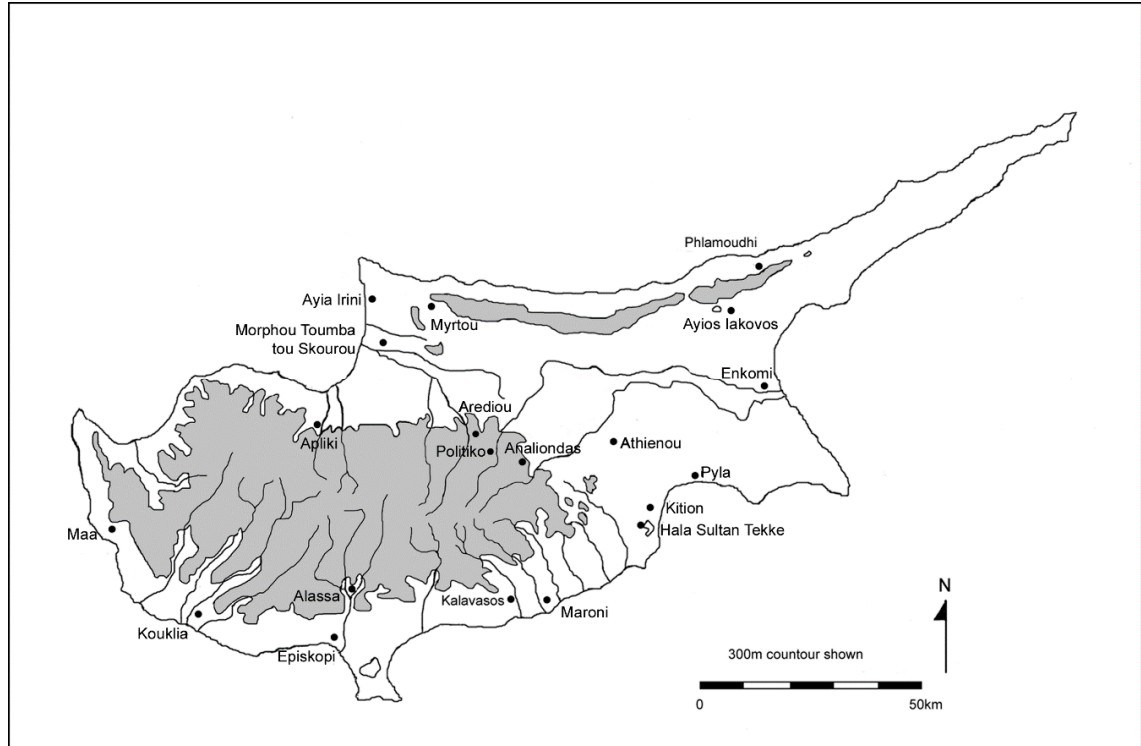

**Figure 1.** Map of Late Bronze Age Cyprus, indicating key sites.

## 3. Materiality, the New Materialisms and Agency

The new materialisms [1,2,5,9,10] question human exceptionalist ontologies and challenge notions that matter plays an incidental role in the construction of people's lives. This approach, essentially a relational ontology [2,3,5], emphasizes that people are not separate and distinct from the rest of the material world, nor are they simply entangled within it. Instead, people are part of the matter of the world and as such co-produce it with other materials and substances. This paper focuses specifically on materiality. As noted by Ingold [24] (p. 27), materiality has proven to be a problematic term, one with "two faces". It is more than simply a description of the material culture of a people; at one level the term simply refers to the physical properties and capacities, indeed the very thingness, of objects [24–26], in effect emphasizing materials and substances (an understanding that is definitely in keeping with the new materialisms). In recent archaeological and anthropological parlance, however, materiality has been understood to refer to a recursive relationship between people and things and how these are entangled [24,25], but situated within an anthropocentric understanding of the world in which people ascribe meaning to their object world. The following discussion returns to the first definition,

namely, the thingness of things, and explores how the capacities of different substances, including the matter of humans, are co-constitutive. As Attala and Steel note (p. xviii) [2], "bringing materials to the foreground not only shows that materials are instrumental in providing the character and meaning of an item, but also that the materials themselves are determining—even actively responsible—for the final shape and manner by which the finished article can manifest." Thus, the materiality, or physical properties, of a substance determines how it behaves, and thus how it might be used and manipulated by people [27]. This focus on physical properties and capacities recognizes the agency of matter.

The new materialisms contend that agency is not specific to a rational (human) subject [1]. We might view matter as an actant, a source of action, noting that it can do things and can produce effects [28], akin to Gell's causal agency [29]. The approach employed in this paper, however, moves beyond this anthropocentric understanding of agency and instead draws specifically on thing-power, which Bennett describes as "[t]he curious ability of inanimate things to animate, act, to produce effects dramatic and subtle" (p. 6). Focusing on the materiality and agency of matter provides us with new ways of thinking about people's interactions with the material world, and indeed draws attention to the transformative role that matter plays in the creation of material and social worlds [18]. As Bennett comments (p. 60), to discern the vitality of a substance enables us "to collaborate more productively with it" [10]. The new materialisms, therefore, recognize the co-creative dynamics of materials and firmly situates humans as part of these matterings.

## 4. The Matter of Late Bronze Age Cyprus

The LC period was characterized by an explosion in material engagements, for the most part experienced by the inhabitants of the urban communities, although it also reached into the rural hinterland. In addition to the daily household praxis shaped by ceramics, ground stone and textiles [30–32], there is evidence for increasingly elaborate objects made from bronze; gold; silver; ivory; glass; faience; and stones such as haematite, lapis lazuli and chlorite [33,34]; some objects were crafted locally, while others were imported to the island. It was not simply a greater range of materials being used in households and deposited in tombs, but likewise the sheer abundance of new object-types available for consumption. In many ways, given the emphasis of this Special Issue on the Bronze Age, and indeed the pivotal role Cypriot copper played in the island's maritime trading fortunes during the second millennium, it might make sense to explore the increasingly complex world of copper production [35–37]. Certainly, metals lend themselves to a thing-power approach. Bennett (p. 60), for example, draws attention to the close relationship between metallurgists and the metals they work with. These craftsmen are intimately aware of the properties of their chosen metal and how it interacts with other substances (alloys, water and fire) and, rather than seeking to impose their will over matter, they desire to see what metal itself can do, appreciating its "shimmering, potentially violent vitality" (p. 61) [10]. Nonetheless, perhaps the most characteristic and widely experienced aspect of LC material culture was its pottery, which was not only embedded in daily household interactions [30] but was widely exported around the East Mediterranean. I would argue that our own engagement with LC ceramic objects might allow us to explore the material experiences of the communities who made and used these objects; thinking about them from the perspective of thing-power will likewise allow us to explore how the capacities of clay shaped and informed the LC social and material world.

## 5. The Vibrant Matter of Clay

Clay/earth holds a special place in human–nonhuman relationships, not only as one of the first substances to be physically transformed from a malleable to durable state as far back as the Upper Palaeolithic [38], but more consistently from the Neolithic [24], but also as one of several mineral substances that in many societies is perceived to be animate and imbued with "a spiritual energy and life-force" (p. 2) [39]. As a substance, clay is very malleable, allowing it to be shaped, formed and moulded into a multiplicity of forms, but the transformation of clay into ceramic depends upon an alchemy of material interactions. In *La Potière Jalouse*, Lévi-Strauss [40] defined the three basic elements

of clay, fire and water that are needed to make pottery. To these we might add a number of other material interactions: some clays are very fine and to increase their tensile strength need to be mixed with other matter—chaff or temper—while other clays are naturally gritty and might need to be refined or washed.

Making pottery is an essentially haptic process, in which the potter works with clay and responds to its physical properties. During this process, the boundaries between the matter of the potter's hands and the substance s/he manipulates are blurred, permeable and fluid. Malafouris extends this further to blur the boundaries between the mind of the potter and matter that s/he shapes, noting (p. 9) that "minds and things are continuous and interdefinable processes rather than isolated" [41] and concluding that mind, body and substance have equal agency within the relationships between clay and the cognitive and bodily skills needed to transform it into vessels. Once fired, this substance becomes durable, indeed virtually indestructible, depending on the heat of the firing, but this process is not always easy to achieve, depending upon control over the fire's temperature, and various examples of misfired, sometimes even vitrified, sherds and wasters are common at pottery workshops, such as the LC production site identified at Sanidha *Moutti tou Ayiou Serkou* [42].

## 6. Animating Clay in Late Bronze Age Cyprus

How then did people work together with clay, water and fire to co-produce the LC material world? The malleability of the clay, mixed with water, allowed for rapid production of numerous items of material culture, which are found abundantly in all LC settlements. These primarily comprise pottery [43,44] and figurines [45], both human and animal. Other typical ceramic objects include lamps and the enigmatic wall bracket [46,47], the function of which remains elusive. One of the more intriguing aspects of LC ceramic production is the sheer diversity of wares in circulation, both fine tableware and utilitarian wares involved in food storage, preparation and cooking [30]. During the earlier part of the LC period, the choice of ware was subject to regional variation but by the 14th century BCE this was largely standardized throughout the island [30,43,48]. There was also an intriguing tension between handmade and wheelmade pottery [43,44,48]; despite the ability to throw pottery on the wheel, the main tableware used throughout the island (White Slip, Base Ring and Monochrome) was high-quality and very desirable handmade wares. This tension subverts traditional accounts of pottery production, which have tended to view wheelmade pottery as technologically superior [43], ignoring the skilled crafting and both the technological and aesthetic achievement of handmade ceramics. In the case of Base Ring ware, I would contend that the choice to produce this pottery by hand (a crafting tradition that persisted for some 500 years) was in fact a direct product of the thing-power of the clays used to make this ware, namely, their plastic properties, as well as the inviting tactile feel and visual impact of the finished, fired product.

This discussion focuses on the Base Ring ware (Figure 2), which was not only experienced by Cypriot householders throughout the Late Bronze Age but was also widely exported to the Levant and Egypt [34]; moreover, this was the fabric type used to fashion the LC ceramic figured world—bovine vases and figurines, and anthropomorphic figurines (Figures 3 and 4a,b) [45]. This ware was integral to the creation of the LC social and material worlds in the coastal towns and the rural communities of the hinterland. As Vaughan (p. 86) observes the "technical ceramic standards manifested by a significant quantity of Base Ring Ware represented a remarkable achievement by potters" [49]. Vaughan's detailed petrographic study of Base Ring [50] demonstrates how the LC potter learned to work with very specific pastes to produce some remarkable vessels. Base Ring pottery is characterized by thin, hard-fired walls. It was fired at a high temperature and sometimes double-fired. It was formed from a very plastic clay, which did not have the tensile strength to be thrown on the wheel, but which potters learned to hand-build into a variety of forms used for pouring and consuming liquids, including delicate deep cups; carinated cups; jugs and juglets; and occasional deep kraters, such as that found in the palace at Alalakh [51], and the bull rhyton (Figure 3). Particularly impressive is the group of jugs with

exaggerated spouts from Kalavasos *Ayios Dhimitrios* Tomb 21, standing between 30 and 50 cm in height but with walls only 2–4 mm thick.

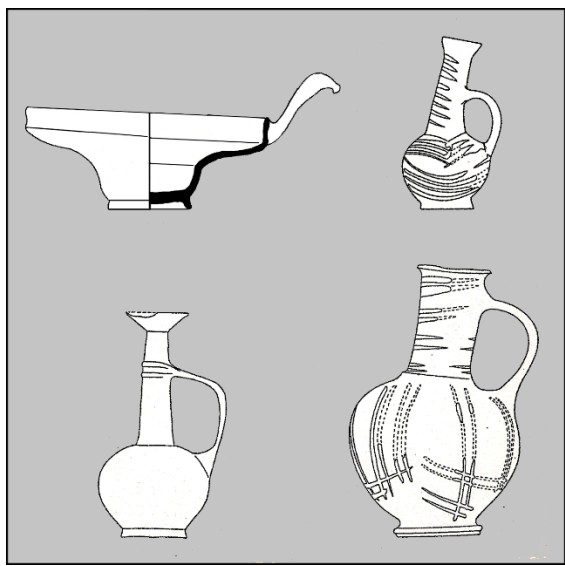

**Figure 2.** Base Ring pottery, after Steel 2004, Figures 6 and 7.

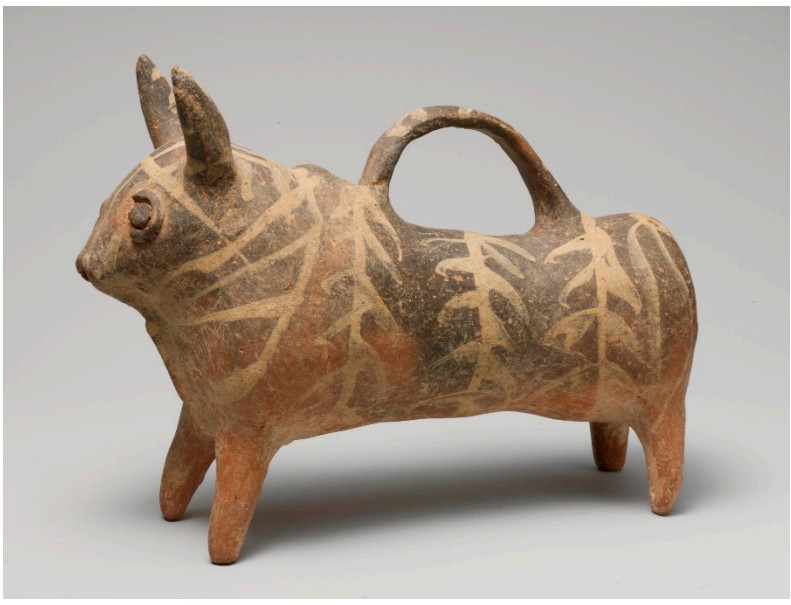

**Figure 3.** Base Ring "bull" rhyton, Cesnola Collection, 74.51.826. Courtesy of the Metropolitan Museum of Art, New York.

Inevitably, as demand for Base Ring tableware increased, by LC II the potters chose to use a more robust paste with a coarse grain, albeit still unsuited for throwing on the wheel, which allowed them to rapidly form the requisite vessels. Alongside this, there was a remarkable standardization of form, contracting to simple small juglets and carinated cups [43]. Moreover, while the earlier vessels were slipped and had a very lustrous or metallic finish [50], the later products of the workshops were either left unslipped or were dipped in a matte slip [43]. The earlier vessels might have relief or incised relief decoration, typically wavy lines or spirals, while some of the later matte-slipped vases might have linear white painted decoration. The apparent decline in production in LC II reflects the properties and capacities of the new clays and slips and demonstrates an ongoing dialogue between materials

and potters, as the latter responded to changing social worlds and the needs of mass production. Intriguingly, despite the need to rapidly produce their vessels, the preference for clays more appropriate for handmade production persisted.

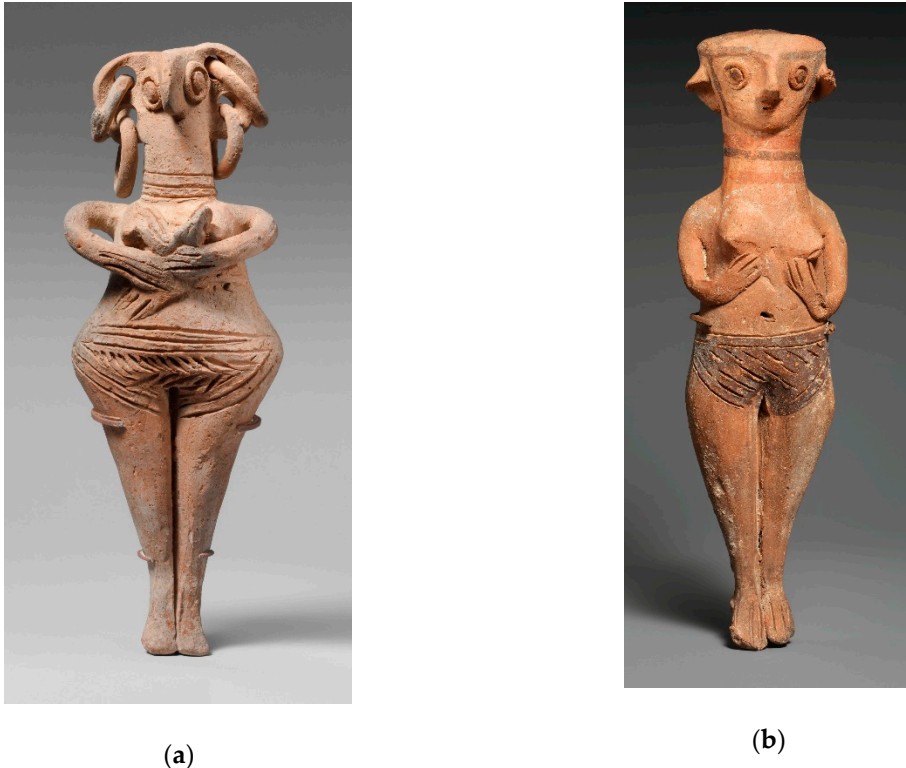

(**a**)                     (**b**)

**Figure 4.** (**a**) Base Ring earring/bird-faced figurine, Cesnola Collection, 74.51.1542. Courtesy of the Metropolitan Museum of Art, New York; (**b**). Base Ring flat-headed figurine, Cesnola Collection, 74.51.1549. Courtesy of the Metropolitan Museum of Art, New York.

The "bull" rhyton [52,53] (Figure 3), introduced in LC IIA, stylistically is an extension of Base Ring pottery production; these vessels have similar surface treatment, including slip and white-painted linear decoration and were undoubtedly made in the Base Ring workshops. These are sturdy little vases standing on four short legs, with a cylindrical body and shoulder hump, handle attached to the back and a modelled tail. The head is realistically modelled with relief pellet eyes, modelled ears, and horns; there is an aperture in the nostril for pouring liquid, and there is an aperture on the back, placed in front of the handle, for filling the vase. Closely related to this form are Base Ring bovine figurines (differentiated by the absence of spout, pouring hole and handle). Two types of female figurine were likewise fashioned from the Base Ring pastes from LC IIA: the bird-faced or earring figurines (Figure 4a) and the flat headed figurines (Figure 4b) [45]. In contrast to the bovine vases and figurines, the surface treatment of these female figurines was very different, being unslipped and with a typically light buff surface colour. The facial and body details were primarily modelled in clay, with relief pellet eyes, pinched nose, and modelled ears and breasts, but with incisions marking the pubic triangle. The flat headed figurines had painted bands (matte red and black paint) around the neck and dark washy paint over the pubic triangle. The skills involved in knowing where to procure and how to work with the Base Ring pastes to form these figures indicate that these were produced by skilled potters, which is further substantiated by Knox's observation [45] (p. 168, Figure 66) that the attachment of the legs and arms used the same techniques as the attachment of handles of pottery vessels, which were thrust through the body.

Base Ring pottery sheds light on the very tactile nature of ceramic production in Late Bronze Age Cyprus; it evidences the ways in which LC potters worked with a very specific clay (a very plastic, fine-grained substance) and responded to its physical properties, which behaved in particular ways when formed by hand but did not respond to the wheel [34,43]. The skills of the pottery forms and the liveliness of the figured pieces demonstrates the animation of this clay: the potters were intimately aware of what these materials could do and actively collaborated with them to produce a distinctively LC material world. The manual shaping of these pastes suggests a close correspondence, indeed blurring of boundaries, between mind, hands and earthy matter, forming what, to paraphrase Bennett (p. 53) [8], might be described as an agentic assemblage, a coming together of conative bodies to co-produce the material world. While we might acknowledge the potter's expertise and confident manipulation of the clay and their skills in firing at high temperatures, we should equally recognize that the capacities of the clay informed many of the choices made within the *chaîne opératoire*, from the initial selection of raw materials (clay, water and temper), to the forming of the vessel/figurine, through to the final firing. Other properties of the finished Base Ring reveal its appeal not just in Cyprus, but more widely throughout the Levant and Egypt [34]: the smooth lustrous surface of earlier pieces was inviting to the touch; the range of surface finishes from lustrous, through metallic slips, to mottled matte surfaces, sometimes with painted decoration, all had visual impact, in particular against the plain wares typical of the region in the second millennium [54].

Several anthropological studies have demonstrated the blurring of boundaries between people and other materials and substances [2]. Rahmen [55], working with the Xié of Amazonia, shows how the body is shaped through co-generative interactions of water, tobacco and flesh, while in rural Sri Lanka, Lamb [56] describes how personhood seeps beyond the boundaries of the human body, soaking into and becoming one with the landscape. Considering the material engagements of Base Ring vessels from an agential realist perspective as relational phenomena [4,5] draws attention to such co-mingling and porosity of matter in Late Bronze Age Cyprus. The wines, oils and other liquids contained in the vessels would certainly have seeped through and become one with the fabric of jugs and juglets. These liquids were intended to be consumed by people, either ingested and internalized or anointed on skin, hair and textiles, becoming-with the body. The anthropomorphic figurines were suited to be hand-held, perhaps suggesting an intimate bodily engagement between object and person within certain ritualized performances, obscuring the boundaries between mind–body–object. This approach recognizes the clay objects, their contents and the people who made and used them as materials in relationship, and illustrates how through these embodied practices Base Ring co-produced the human body.

## 7. Conclusions

This paper explores material entanglements in Late Bronze Age Cyprus, focusing on clay and Base Ring pottery/figurines. The aim has been to demonstrate the unique ways in which clay and the potter/figurine-maker co-produced the material world. Bennett's thing-power [10] demonstrates that we should not privilege human agency over the conative properties of clay and other seemingly inert substances. Instead, focusing on the vitality of matter allows us to think about material transformations involved in pottery production as the coalescing of human and material agency and indeed blurs the boundaries between them. While the potter moulded and formed the clay, these actions were both constrained and enabled by its properties—essentially, as highlighted by Malafouris [41], there is no distinction between human and material agency. Base Ring pottery and figurines emerged from an inevitable tension of mediated activity between substances and people. Similarly, thinking about the Base Ring vessels, the substances they contained and the people who consumed them as things-in-phenomena further emphasizes the fluidity, porosity and relationality of the material world, including the human body, something which is increasingly evident in ethnographies [2,55,56] but remains obscure within archaeological narratives.

**Funding:** This research received no external funding.

**Conflicts of Interest:** The author declares no conflict of interest.

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
