# Peer review of "Feats of Clay: Considering the Materiality of Late Bronze Age Cyprus"

_sustainability, doi:10.3390/su12176942_

Round 1
Reviewer 1 Report
This is a nice little contribution based on new materialism approaches.
Some corrections with regard to technicalitities should be done, e.g.,
to systmatize the bibliography concerning italics (see and compare, e.g.,
21, 24, 25, and 26; 36 and 37; etc.)
And, line 68: "might be use" to "might be used."
Author Response
Thank you for your positive response to my article. I have made the corrections (typo line 68 and formatting of bibliography - italics) as highlighted in your review.
Reviewer 2 Report
This is a wonderful and well-written paper, which I have no hesitation recommending for publication.
It discusses the role of clay, not just as an extremely malleable material, but also as shaping human lives and choices in a multitude of ways. As such, it as a good example that I would potentially include in classes on materiality in archaeological theory.
Below only some very minor comments and suggestions:
- Introduction: repetition of ‘explores’/‘exploring’.
- 125: possibly use ‘they’ instead of s/he
- 124-125: “various of misfired”, word missing?
- 147 and following: “bull vases, bull figurines”. This is indeed the standard terminology, but since they are actually not all bulls, maybe “bovine” would be better, or “bull” in quotation marks
- Fig. 3: difficult to tell for certain, but there seems to be two openings, both at the shoulders, and through the muzzle, which would make it a rhyton
- 161: “cups Earlier”, punctuation missing
- 173-174: These are certainly rhyta (cf. Koehl 2006, Bronze Age Aegean Rhyta)
Author Response
Thank you for your very positive endorsement of the article. Thank you for highlighting the typos and repetitions, which are addressed accordingly:
12: changed "explores" to "examines"
118: I kept s/he rather than changing to they to keep open the question of gender of the potter, rather than obscuring this.
124-5: various examples of misfired
147: "bull vases, bull figurines" changed to bovine vases and figurines"; 166: Figure 3 Base Ring "bull" rhyton; 168: "bull" askos; 177: bovine vases and figurines
161: full stop after cups
173-4 (back to 168): changed askos to rhyton and added reference to Koehl.
Reviewer 3 Report
This is a very strong paper. It provides a good overview of the theory with a very well worked and well explained example. The author presents the theoretical material succinctly and then lets the material (literally!) do the talking as she discusses the nature of specific clays, their properties and how this shaped the emergence of a specific material culture repertoire. My comments below are very minor as the paper is already excellent.
A few more citations to the growing literature within archaeology on new materialisms and assemblage theory are probably appropriate to situate the piece within the broader literature on the topic e.g. Jones 2012, Jervis 2018; Jones and Hamilakis special in CAJ etc.
I wonder whether in section 3 you might want to reconsider the discussion of materiality. The new materialisms move beyond materiality approaches to place people firmly within the material world as you so very effectively highlight. Bringing in the concept of materiality here has the potential perhaps to confuse some readers particularly given, as you note, the “two faces” where on the one hand Miller et al really focus on the effect of things on people in a rather anthropocentric manner and Ingold really focuses on properties in a way that is more akin to new materialism. My question is whether avoiding the term materiality in the introduction and instead sticking to just talking about new materialism and perhaps avoiding the discussion of materiality in section 3 could potentially avoid some confusion.
In section 4 where you mention the potential of metal to be discussed in a new materialist frame you could cite Crellin 2020 with its extended case study looking at Bronze Age metalwork using new materialism.
I wonder whether there is potential to bring out a little more about the specific clay used to make the base-ring vessels – in traditional accounts of pottery making wheel thrown is seen as ‘technologically more complex’ and often ‘superior’ to handmade. In your example we can see the fallacy of this approach to technology that presumes increasing mechanism is both superior and more complexity – the specific properties of the clay mean hand-making is best and it allows more exciting and complex designs. You say much of this already but I wonder about pulling this out a touch more
Author Response
Thank you for your very positive feedback on this article and in particular the comments on how to strengthen and/or clarify the theoretical discussion, which are very help.
- I have added in references to the growing literature on NM and assemblages in archaeology. line 28: references to Harris 2014, Lucas 2017, Jervis 2018;
- Discussion of materiality: I think it is important to leave this in as there is a certain ambiguity in terminologies in the literature, but I have attempted to clarify the distinction between discussions of antrhopocetnric concepts of materiality and the NM perspective (lines 57-66) by slightly reframing the discussion.
- Section 4: potential of metal in NM analysis. I have added reference to Crellin 2020, Change and Archaeology.
- I have added to the discussion on handmade pottery and the technological superiority debate: lines 143-9 and 168-179.